# The Effects of ZnTe:Cu Back Contact on the Performance of CdTe Nanocrystal Solar Cells with Inverted Structure

**DOI:** 10.3390/nano9040626

**Published:** 2019-04-17

**Authors:** Bingchang Chen, Junhong Liu, Zexin Cai, Ao Xu, Xiaolin Liu, Zhitao Rong, Donghuan Qin, Wei Xu, Lintao Hou, Quanbin Liang

**Affiliations:** 1School of Materials Science and Engineering, South China University of Technology, Guangzhou 510640, China; ChenBC_999@163.com (B.C.); jhliu1@163.com (J.L.); 13005112535@163.com (Z.C.); m13856403915@163.com (A.X.); liulin9708@163.com (X.L.); rzt1512388848@163.com (Z.R.); l.quanbin@mail.scut.edu.cn (Q.L.); 2Guangdong Provincial Key Laboratory of Optical Fiber Sensing and Communications, Guangzhou Key Laboratory of Vacuum Coating Technologies and New Energy Materials, Siyuan Laboratory, Department of Physics, Jinan University, Guangzhou 510632, China; 3Institute of Polymer Optoelectronic Materials & Devices, State Key Laboratory of Luminescent Materials & Devices, South China University of Technology, Guangzhou 510640, China

**Keywords:** nanocrystal, CdTe, Cu-doped, ZnTe, solar cells, solution processed

## Abstract

CdTe nanocrystal (NC) solar cells have received much attention in recent years due to their low cost and environmentally friendly fabrication process. Nowadays, the back contact is still the key issue for further improving device performance. It is well known that, in the case of CdTe thin-film solar cells prepared with the close-spaced sublimation (CSS) method, Cu-doped CdTe can drastically decrease the series resistance of CdTe solar cells and result in high device performance. However, there are still few reports on solution-processed CdTe NC solar cells with Cu-doped back contact. In this work, ZnTe:Cu or Cu:Au back contact layer (buffer layer) was deposited on the CdTe NC thin film by thermal evaporation and devices with inverted structure of ITO/ZnO/CdSe/CdTe/ZnTe:Cu (or Cu)/Au were fabricated and investigated. It was found that, comparing to an Au or Cu:Au device, the incorporation of ZnTe:Cu as a back contact layer can improve the open circuit voltage (*V*_oc_) and fill factor (FF) due to an optimized band alignment, which results in enhanced power conversion efficiency (PCE). By carefully optimizing the treatment of the ZnTe:Cu film (altering the film thickness and annealing temperature), an excellent PCE of 6.38% was obtained, which showed a 21.06% improvement compared with a device without ZnTe:Cu layer (with a device structure of ITO/ZnO/CdSe/CdTe/Au).

## 1. Introduction

CdTe is a II–VI group of semiconductor materials with a moderate band gap of 1.45 eV and a high optical-adsorption coefficient (over 10^4^ cm^−1^ in the optical range), which is promising for light harvesting, and regarded as an attractive material for solar-cell applications [1,2,3]. One of the major problems for CdTe solar cells is the difficulty in obtaining low and stable ohmic contact to CdTe (the low resistance of CdTe/back contact electrode). It is noted that CdTe has a very high electron affinity (x = 4.5 eV) and low carrier concentration (~10^14^/cm^3^). There are almost no metals with such a high work function to form ohmic contact to CdTe [4]. In order to obtain low-resistance contacts to CdTe, there are several ways to establish an interface that provides suitable electrical properties. The first uses interlayer contact materials with a high work function, such as MoOx [5,6], WOx [7], CuSCN [8], and cobalt phthalocyanine [9] before electrode deposition. Formation of a heavily doped region at the surface of a CdTe thin film is also an important way to reduce the barriers for hole-collecting. Cu is the most commonly used metal to create acceptor states with concentrations as high as ~10^19^/cm^3^ in CdTe [10,11,12]. An investigation shows that Cu exists as Cu_i_^+^ and forms shallow donor states or deep acceptor states Cu_Cd_^−^ in CdTe [13,14]. It was found that copper atoms can move quickly in a cadmium telluride thin film after heat treatment [15]. Excess-doped Cu leads to Cu gathering at the interface of the p-n junction and forming an *N*-type compensation caused by Cu^+^, which leads to a decline in device performance [16,17]. Therefore, controlling the effective doping of copper is the key to obtaining high efficiency and stable thin-film solar cells. In order to increase stability, the chemical etching of the cadmium telluride surface is essential prior to Cu deposition in order to create a Te-rich surface [18,19]. The used etching solution includes a nitric acid solution and a bromo methanol solution. The tellurium-rich layer formed on the surface can combine with the deposited copper to form Cu_x_Te to prevent the excessive diffusion of copper. Another way to control the Cu doped in CdTe is by using ZnTe:Cu as a back contact layer. As the ZnTe has a similar lattice parameter and valence band as that of CdTe, it is an ideal back-contact material for CdTe solar cells as the well-matched interface, which works as an electron blocking layer [20,21]. On the other hand, high-acceptor-level doping of ZnTe (>10^20^/cm^3^) can be easily obtained by varying the Cu-doped content, which allows for the formation of an ideal tunnel junction to the contact-metal layer [22].

Although back-contact techniques have been well developed for CdTe-based thin-film solar cells fabricated by vacuum technology, there are still few reports on solution-processed CdTe nanocrystal- (NC) based solar cells [23,24,25]. There are many merits of solution processed CdTe NC solar cells, for example, low materials consumed, low cost and simple fabricating techniques. As much of the grain boundary existed in the CdTe NC layer (CdTe grain size is about 100–200 nm), common corrosion technology cannot obtain a Te-rich surface, as corrosion solvents quickly diffuse along the grain boundary and make device decay or shunting, which has been confirmed in previous work [26]. Inserting a hole-transport layer with high work function before the deposition of an electrode metal contact has been widely accepted for improving the ohmic contact for CdTe solar cells [9,27]. Recently, inspired by the use of MoO_x_ in organic solar cells, MoO_x_ was developed as hole transfer layer for CdTe NC solar cells, and improved efficiency is found in this case [28]. Following this, organic hole-transport materials with a high work function were also developed as the back contact layer, with the advantage of low-cost solution processing. For example, Spiro-OMeTAD (2,2,7,7-tetrakis(*N*,*N*-di-4-methoxyphenylamino)-9,9-spirobitluorene) [29] and P3KT [30] are used as a back contact layer for reducing interface recombination and improving band alignment, so drastic improvement in device performance is obtained. Most recently, a new cross-linkable conjugated interface polymer TPA was successfully applied in the CdTe NC-based solar cells, and power conversion efficiency (PCE) as high as 8.34% was obtained, which is the highest value ever reported for any solution processed CdTe NCs solar cells with an inverted structure [31]. In the previous report, ZnTe:Cu contact was first applied in CdTe NC solar cells with the configuration of FTO/SnO_2_/CdS/CdTe/ZnTe:Cu/Ti [32]. Although good ohmic contact was obtained in this case, the short circuit current density *J_sc_* was low due to low junction quality. In previous reports [33,34,35,36], it was found that high junction quality is expected when a commonly used *n*-type partner (such as CBD-CdS [37] or TiO_2_ [38,39] prepared by precursor decomposition etc.) is replaced by solution-processed CdSe or CdS NC due to the similar size and structure as CdTe NCs, which significantly reduced interface recombination and resulted in high device performance. Herein, we fabricated CdTe NC solar cells via solution-processed CdTe and CdSe NC as donor/acceptor materials with the configuration of ITO/ZnO/CdSe/CdTe/back contact/Au. Cu-doped ZnTe and Cu:Au were selected as back contact, while a device without any back contact was also fabricated and investigated. A PCE of 6.38% was observed from the device with ZnTe:Cu as back contact layer.

## 2. Experiments

CdTe, CdSe NCs, and the Zn^2+^ precursor were prepared according to the previous reported method [33]. The CdSe and CdTe NC films were deposited using layer-by-layer spin-coating and sintering under ambient conditions. A typical fabrication process is presented in the following. The ZnO thin film was prepared by spin-casting Zn^2+^ precursor on the ITO substrate and annealing at 400 °C for 10 min. Several drops of CdSe NC solution (a mixture of pyridine and 1-propanol with a volume ratio of 1:1 at 30 mg/mL) were put on top of the ITO/ZnO substrate and spin-casted at 3000 rpm for 20 s. Then the substrate was placed on a hotplate at 150 °C for 10 min and transferred to another hotplate at 350 °C for 40 s. This process was repeated two times and the CdSe NC film thickness was about 80 nm (40 nm per layer). Following this, several drops of CdTe NC solution (with 45 mg/mL in pyridine and 1-propanol with a volume ratio of 1:1) were deposited on the ITO/ZnO/CdSe thin film and spin-casted at 1100 rpm for 20 s. The ITO/ZnO/CdSe/CdTe samples were then placed at a hot place at 150 °C for 3 min. Then the substrate was dipped into saturated CdCl_2_/CH_3_OH solution for 10 s and rinsed in 1-propanol. Then the substrate was placed immediately on a hotplate at 350 °C for 40 s (~100 nm each layer). This process was repeated for five times in order to obtain an optimized active layer thickness. The thickness of ZnO, CdSe and CdTe were ~40 nm, ~80 nm and ~500 nm respectively. Detail process can be found in the literature [31]. After washing and cleaning, ZnTe films were deposited onto the ITO/ZnO/CdSe/CdTe substrate via a thermal evaporation process at a rate of 8 Å/s through a shadow mask with an active area of 0.16 cm^2^ under a vacuum pressure of 4 × 10^−4^ Pa. Following this, Cu and Au (60 nm) were deposited on the ZnTe through a shadow mask by thermal evaporation. The ITO/ZnO/CdSe/CdTe/ZnTe:Cu/Au and ITO/ZnO/CdSe/CdTe/Cu/Au thin films were then placed on a hot place and annealed at different temperatures. The area of all the NC solar cells is 0.16 cm^2^. The atomic force microscopy (AFM) imagines were obtained using a NanoScope NS3A system (Veeko, CA, USA). The morphology and structure were further characterized by scanning electron microscopy (SEM, Nova NanoSEM430, Thermo Fisher Scientific, Eindhoven, The Netherlands) and X-ray diffraction (XRD, X’pert Pro M, Philips, Amsterdam, The Netherlands). The external quantum efficiency (EQE) was measured by a Zolix instrument (Solar Cell Scan100, Zolix Instruments Co., Ltd., Beijing, China). The *J*–*V* characteristics were measured with a Keithley 2400 under an illumination of 100 mW/cm^2^ with an air mass 1.5 (AM 1.5) solar simulator (Oriel model 91192). The transient photovoltage measurements (TPV) were taken out by using the OmniFluo system (Zolix, Beijing, China).

## 3. Results and Discussion

To investigate the effect of ZnTe on the surface morphology of the CdTe NC layer, the morphology of ITO/ZnO/CdSe/CdTe/ZnTe with different ZnTe thickness was characterized by atomic force microscopy (AFM). The ZnTe film was deposited on the CdTe NC layer by thermal evaporation through a shadow mask and annealed at 200 °C for 30 min. As shown in Figure 1a, without ZnTe, the surface of the CdTe NC showed a uniform and compact structure, with a root-mean-square (RMS) roughness of 8.26 nm. It was clear that, with the ZnTe thickness increase from 10 nm to 50 nm, grain size increased linearly. The RMS values for ZnTe with thickness of 10 nm, 20 nm, and 50 nm were 8.93 nm, 7.62 nm, and 10.8 nm, respectively. There were many small particles for the 10 nm ZnTe sample (Figure 1b), while a compact and uniform surface was found for the 20 nm and 50 nm ZnTe samples. The smooth and compact surface obtained in the ZnTe sample implied good physical contact between CdTe and ZnTe, which was essential to decrease the interfacial contact resistance and improve the fill factor of the solar-cell device.

The structure and composition of ZnTe were further characterized by X-ray diffraction (XRD) and energy dispersive spectrometry (EDS). The XRD sample was prepared by depositing 100 nm on the Si substrate and then annealing at different temperatures for 30 min. As shown in Figure 2a, diffraction patterns with peaks at about 25.2°, 29.3°, 41.87°, 49.6°, 51.8°, 60.7°, 66.8°,68.9°, 76.6°, and 81.9° were identified from the XRD pattern, corresponding to the (111), (200), (220), (311), (222), (400), (331), (420), (422), and (511) planes, respectively, of the ZnTe zinc blend structure. It is noted that pure ZnTe peaks were found for samples without annealing. On the contrary, there were many peaks corresponding to the pure Te element that emerged when the samples were annealed at a temperature of up to 300 °C, which may be due to the decomposition of ZnTe at high temperatures. From the energy dispersive spectrum (Figure 2b), the relative amount of Zn to Te in the sample was close to 1:1, illuminating the formation of a ZnTe alloy. The presence of Si and O is attributed to the SiO_2_ substrate used for the deposition of the ZnTe film.

The typical device architecture of CdTe NC solar cells with a ZnTe:Cu back contact layer is presented in Figure 3a. The ZnO thin film prepared by decomposition of the Zn carboxyl precursor was selected as the electron-transfer layer and prevented from casting shunt during the deposition of the active CdTe/CdSe layer. The active CdTe/CdSe layer was prepared with a layer-by-layer process. Prior to ZnTe deposition, the CdTe NC thin film was cleaned by ultrasound in methanol several times. Figure 3b shows the band alignment of ITO, ZnO, CdSe, CdTe, ZnTe:Cu, and Au. The introduction of ZnTe between CdTe and the anode optimizes energy-level alignment and decreases contact resistance, which improves charge collection and reduces interfacial carrier recombination. The surface and cross-section images of ITO/ZnO/CdSe/CdTe/ZnTe:Cu/Au were investigated using scanning electron microscopy. From Figure 3c, it is evident that CdTe grain size after chemical treatment/annealing was larger than 100 nm, and the whole film was compact and smooth, which is preferable for efficient carrier collecting due to reduced grain boundaries and interface defects. The cross-sectional SEM image of CdTe NC solar cells was included in Figure 3d. Active-layer (CdTe/CdSe) thickness was ~600 nm, prepared by depositing two layers of CdSe and five layers of CdTe NC onto ITO/ZnO in sequence.

It is well known that the most efficient CdTe thin-film solar cells are based on Cu: Au back contact [40,41]. CdTe NC solar cells with a configuration of ITO/ZnO/CdSe/CdTe/Cu/Au were fabricated by using different Cu thicknesses. In Appendix A, we show the current density-voltage (*J*–*V*) curves obtained for CdTe NC solar cells with 1.3 nm and 2.1 nm Cu as back contact. The optimized CdTe NC solar cells with 2.1 nm Cu yielded a *V*_oc_ of 0.62 V, a *J*_sc_ of 13.09 mA/cm^2^, an FF of 55.84%, leading to a PCE of 4.53%, while these values were 0.53 V, 8.91 mA/cm^2^, 49.12%, 2.32%, for the 1.3 nm Cu device. The effects of annealing temperature on device performance are shown in Appendix A and the solar cells parameters are summarized in Appendix A. PCE increased as annealing temperature increased from room temperature to 200 °C, then dropped down linearly when we further increased the annealing temperature. Therefore, the PCE obtained for solar cells with a Cu/Au back contact was significantly lower than the device without Cu contact, reported before [31], which was mainly attributed to the low obtained *J*_sc_ in this case. We speculate that this was because the grain size of the CdTe NC active layer was only 100 nm, while this value was up to 1 μm for CdTe thin-film solar cells prepared by the CSS method. Therefore, Cu can quickly diffuse in the entire active layer after annealing due to the large grain boundary, which may result in mid-gap recombination and lead to a low-output current. By using the ZnTe buffer layer, Cu diffusion is restricted, and high device performance is expected. In the case of CdTe NC thin-film solar cells, the thickness of the ZnTe and annealing temperature should have significant effects on Cu diffusion and the device’s contact resistance. ZnTe thickness for efficient CdTe thin-film solar cells prepared by the CSS method was about 100 nm. However, the thickness of CdTe NC solar cells was significantly lower than those CdTe solar cells prepared by the CSS method. Furthermore, the surface roughness of the CdTe NC thin film was small. Therefore, the thickness of ZnTe for efficient CdTe NC solar cells may be different from those reported before. To investigate the effect of thickness on device performance, devices with a ZnTe thickness between 10 nm and 100 nm were fabricated by thermally evaporating ZnTe on the ITO/ZnO/CdSe/CdTe substrate. For comparison, a controlled device with Au back contact was also fabricated. Figure 4a presents the *J*–*V* curves of the best CdTe NC solar cells with different back contact under light conditions (The *J–V* curves of CdTe NC with different thickness of ZnTe:Cu and annealing temperature are presented Appendix A), while Table 1 summarizes the solar cells parameters. From the *J*–*V* curves, we can see that the controlled devices (ITO/ZnO/CdSe/CdTe/Au) showed a PCE of 5.27%, *J*_sc_ of 19.97 mA/cm^2^, *V*_oc_ of 0.56 V, and FF of 47.15%. The low FF value implied that large contact resistance existed on the interface of CdTe/Au. In contract, the best device, with a 20 nm ZnTe/1 nm Cu/Au contact, yielded a *J*_sc_ of 19.73 mA/cm^2^, *V*_oc_ of 0.65 V, FF of 49.75%, delivering a high PCE of 6.38%. The observed PCEs from NC solar cells with a ZnTe:Cu/Au back contact were more than 20% higher than those of the controlled device. The improvement in device performance was mainly attributed to the increased in *FF* and *V*_oc_. From the *J–V* curves under dark, the current at the reversed bias from a device with ZnTe:Cu/Au back contact is lower than that from a device without ZnTe:Cu back contact. The low current in reverse bias voltage under dark and improved *R_sh_* imply that the ZnTe:Cu can serve as an electron blocking layer to effectively prevent the leakage currents [42]. As shown in Table 1, when the thickness of ZnTe increases from 10 nm to 100nm, the *V*_oc_ of NC devices remain at ~0.6 V, while the current density increases from 14.36 mA/cm^2^ (10 nm ZnTe) to 19.73 mA/cm^2^ (20 nm ZnTe), then drops down to 14.40 mA/cm^2^ (100 nm ZnTe). The changes in the fill factor have a similar behavior as that of *J*_sc_ or *V*_oc_. On the other hand, when the thickness of ZnTe:Cu is fixed, the annealing temperature has significant effects on the NC solar cells performance. The best device is obtained at an annealing temperature of 200 °C. We speculate that the interface defects of CdTe/ZnTe:Cu and the diffusion of Cu dominate the device performance. At an annealing temperature of 200 °C and thickness of 20 nm, the interface defects is low and the diffusion of Cu in the ZnTe thin film is homogeneous, which will result in high device performance. On the contrary, too high of an annealing temperature may result in Cu accumulation or inadequate diffusion of Cu in the ZnTe, leading to low device performance.

To investigate the performance improvement in ZnTe:Cu contact NC solar cells, external quantum efficiency (EQE) measurements for the device without a ZnTe:Cu buffer layer were taken and are presented in Figure 5a. It was found that the ZnTe:Cu contact device showed a higher EQE value between 450–650 nm, while it had a lower EQE value between 650–800 nm, when compared to a controlled device. When integrated, the *J*_sc_ was calculated to be 18.41 mA/cm^2^ and 18.70 mA/cm^2^ for device without a ZnTe:Cu contact, which agrees well with the data from the *J–V* curves. As the electron affinity for CdTe and high resistance, there are no metals that can form ohmic contact to CdTe, and Fermi-level pinning was found for all metals [12]. The low EQE response, between 600 nm to 800 nm was attributed to the interfacial recombination of CdTe/Au. When the ZnTe:Cu back contact layer was introduced, the diffusion of Cu in the interface of CdTe and the created acceptor states was at a high concentration [14] and it aligned the Fermi level to the valence band (VB) of CdTe. Then the carriers recombination will be decreased. On the other hand, the dropdown in the EQE value for the ZnTe:Cu device between 650–800 nm may have come from the accumulate of Cu in the CdTe/CdSe junction, which is also found in the CdTe/CdS solar cells prepared by CSS method [43]. It should be pointed out that the *J*_sc_ of the best cell with ZnTe:Cu back contact is still less than that of the control cell in this study. Furthermore, it is noted that the EQE response of devices with a ZnTe:Cu contact was significantly lower than those devices with Si-TPA as back contact, as reported before. We anticipated that, as the diffusion of Cu in ZnTe and CdTe was not homogeneous, few Cu_x_Te could be formed as no Te rich interface. The highly doped *p*^+^ region via wet etching of the CdTe NC surface using dilute bromine/methanol was exclusively faint, as the acid solvent can quickly be transferred from the grain boundary, which was also confirmed by Matthew et al. Therefore, it was difficult to form a Cu_2-x_Te layer at the CdTe interface, and contact resistance was still high, further restricting improvement in the PCE. To further investigate the effects of the ZnTe:Cu buffer layer on the recombination process of NC solar cells, transient photovoltage (TPV) was used to measure the charge recombination in NC solar cells with/without a ZnTe:Cu buffer layer. During the TPV measurement, we obtained a steady-state equilibrium by placing NC solar cells under a white-light bias. By applying another weak laser pulse to NC solar cells, additional charges were generated. The charge recombination of NC solar cells was investigated by tracking the transient voltage associated with perturbations in the charge population. From Figure 5b, we can see that charge-recombination time for NC devices without a ZnTe:Cu buffer layer was 1.00 µs, while this value was 1.67 µs for the device with a ZnTe:Cu buffer layer, which implies the charge-recombination rate was lower in the ZnTe:Cu device compared to devices without a ZnTe:Cu.

## 4. Conclusions

In conclusion, we described a developed ZnTe:Cu back contact layer for CdTe NC solar cells. Comparing it to a device without any back contact layer, we saw that improved solar-cell performance was attained by using ZnTe:Cu as a back contact layer. The improvement in *V*_oc_ and FF could be attributed to a better band alignment and low contact resistance forming on the CdTe surface, which decreased charge recombination in the interface and improved carrier collecting efficiency. A PCE of 6.38% was observed from the NC solar cells with a device structure of ITO/ZnO/CdSe/CdTe/ZnTe:Cu/Au, which was significantly higher than the controlled device with a structure of ITO/ZnO/CdSe/CdTe/Au (5.27%). Our results suggest that the back contact layer recipe involving ZnTe:Cu is applicable to solution-processed efficient CdTe NC solar cells if the ZnTe:Cu film is subjected to more optimized processing.

## Figures and Tables

**Figure 1 nanomaterials-09-00626-f001:**
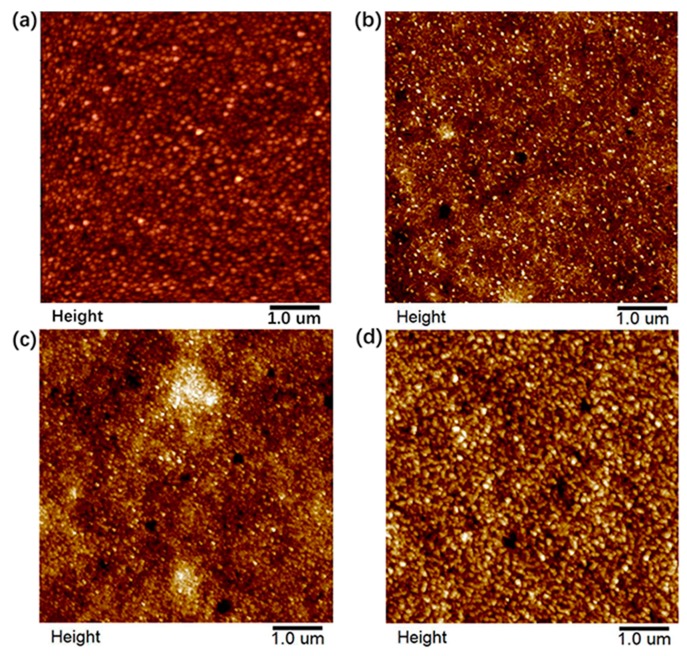
Atomic force microscopy (AFM) images of ITO/ZnO/CdSe/CdTe/ZnTe with different ZnTe thickness. (**a**) Without ZnTe; (**b**) 10 nm ZnTe; (**c**) 20 nm ZnTe; (**d**) 50 nm ZnTe.

**Figure 2 nanomaterials-09-00626-f002:**
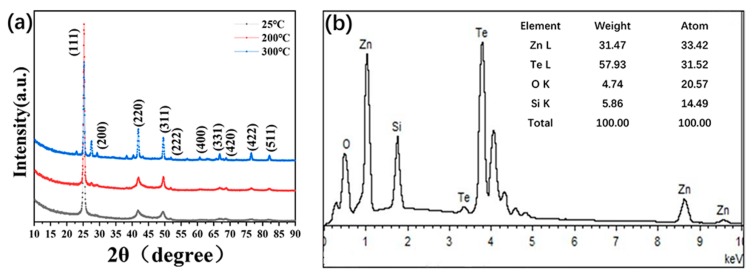
(**a**) X-ray diffraction (XRD) pattern of the ZnTe films at different annealing temperatures; (**b**) Energy dispersive spectrometry (EDS) of the as-prepared ZnTe sample.

**Figure 3 nanomaterials-09-00626-f003:**
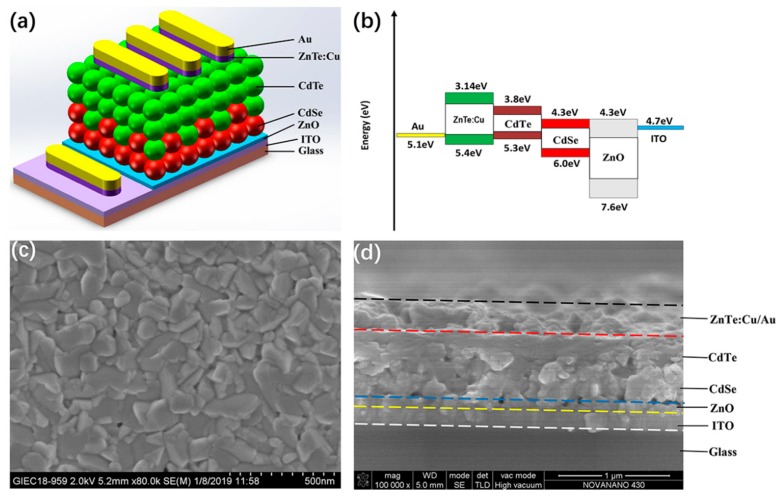
(**a**) Schematic of nanocrystal (NC) solar cells with a structure of ITO/ZnO/CdSe/CdTe/ZnTe:Cu/Au; (**b**) band alignment of ITO, ZnO, CdSe, CdTe, ZnTe:Cu, and Au; (**c**) SEM images of CdTe NC thin film; (**d**) cross-section SEM images of ITO/ZnO/CdSe/CdTe/ZnTe:Cu/Au.

**Figure 4 nanomaterials-09-00626-f004:**
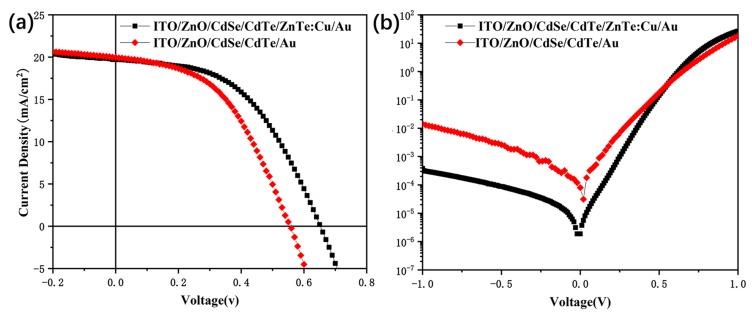
*J–V* curves of ITO/ZnO/CdSe/CdTe/ZnTe:Cu/Au and ITO/ZnO/CdSe/CdTe/Au (**a**) under light and (**b**) under dark conditions.

**Figure 5 nanomaterials-09-00626-f005:**
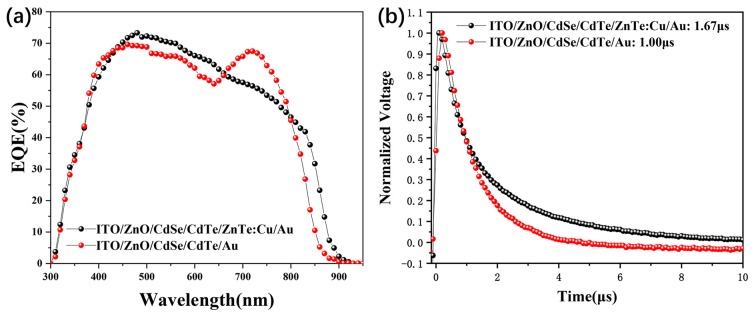
(**a**) External quantum efficiency (EQE) spectrum of a device without a ZnTe:Cu back contact layer; (**b**) transient photovoltage measurements of NC solar cells without a ZnTe:Cu back contact layer.

**Table 1 nanomaterials-09-00626-t001:** Summary of the photovoltaic parameters of the nanocrystal (NC) solar cells prepared under different conditions.

Annealing Temperature (°C)	ZnTe Layer Thickness (nm)	Cu Layer Thickness (nm)	*V*_oc_ (V)	*J*_sc_ (mA/cm^2^)	FF (%)	PCE (%)	*R_s_* (Ω·cm^2^)	*R_sh_* (Ω·cm^2^)
200	10	1	0.57	14.36	45.61	3.73	20.65	387.55
200	30	1	0.63	16.21	50.12	5.12	12.70	425.71
200	50	1	0.62	14.94	48.55	4.5	15.63	380.11
200	100	1	0.59	14.40	41.37	3.51	20.11	250.08
no	20	1	0.49	11.64	32.94	1.88	30.94	111.04
100	20	1	0.59	13.40	40.37	3.19	23.70	213.32
160	20	1	0.60	18.08	44.66	4.84	12.70	245.71
180	20	1	0.64	18.63	47.23	5.63	11.84	346.00
200	20	1	0.65	19.73	49.75	6.38	11.24	349.59
220	20	1	0.61	18.03	48.73	5.36	13.80	201.84
240	20	1	0.60	15.97	45.60	4.37	15.95	381.51
260	20	1	0.61	14.29	41.64	3.63	29.22	261.08
no	0	0	0.56	19.97	47.15	5.27	11.11	209.99

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
