# Peer review of "The Effects of ZnTe:Cu Back Contact on the Performance of CdTe Nanocrystal Solar Cells with Inverted Structure"

_nanomaterials, 2019, doi:10.3390/nano9040626_

Reviewer 1 Report

This paper well described the effect of Cu-doped ZnTe electron blocking layer that grately reduced the reverse saturation current, resulting in the large inrease of open-circuit voltage. This paper is informative and worthwile to be published.

1) Cu-doped ZnTe layer is like a back surface filed in n+pp+ Si juntion. It greatly reduced the recombination current. To me the buffer layer is an electron blocking layer instead of hole transport layer. Anyhow it should be mentioned in the paper.

2) Ater reference 16, please add another reference that explains the degradation mechanims of CdTe solar cells by Cu+ migration from back contact to CdS buffer in addion to Cu agglomeration at the CdS/CdTe interface.  Ref 17. B. T. Ahn, J. H. Yun, E. S. Cha, K. C. Park, Understanding the junction degradation mechanism in CdS/CdTe soalr cells using a Cd-deficient CdTe layer, Current Applied Physics, 12 (2012) 174-178.

Author Response

List of point-by-point changes based on referee’s report

Nanomaterials-465512 

(Italic character- Referee’s comment; red character-authors response to the referee)

Here we submit the revised version of Nanomaterials-465512, in which the comments from the two reviewers are carefully considered and addressed. Our revisions have been highlighted in red color for your easy identification.
Reviewer #1
:
Comments to the Author
This paper well described the effect of Cu-doped ZnTe electron blocking layer that grately reduced the reverse saturation current, resulting in the large inrease of open-circuit voltage. This paper is informative and worthwile to be published.

1. Cu-doped ZnTe layer is like a back surface filed in n+pp+ Si juntion. It greatly reduced the recombination current. To me the buffer layer is an electron blocking layer instead of hole transport layer. Anyhow it should be mentioned in the paper.

Thank for the comments. According to the referee’s suggestion, we have revised the expression in the manuscript.

 2. Ater reference 16, please add another reference that explains the degradation mechanims of CdTe solar cells by Cu+ migration from back contact to CdS buffer in addion to Cu agglomeration at the CdS/CdTe interface.  Ref 17. B. T. Ahn, J. H. Yun, E. S. Cha, K. C. Park, Understanding the junction degradation mechanism in CdS/CdTe soalr cells using a Cd-deficient CdTe layer, Current Applied Physics, 12 (2012) 174-178.

Thanks for the comments. We have added the above reference in the revised manuscript.

Reviewer 2 Report

The authors mainly studied the effects of ZnTe:Cu layers on the performance of CdTe NC solar cells. The title should be revised in order to better fit the main theme of the study. In addition, the cell configuration shown in Fig. 3(a) for this study is superstrate, which is a common term for thin-film solar cells prepared on glass.

1. The abstract should be more concise. The key approaches and the critical results should be presented.

2. Line 24 and 60, the ZnTe:Cu layers work as the back contacts. In line 96, the ZnTe:Cu layers are regarded as the HTL. In line 33, is the ZnTe:Cu layer also the buffer? Clearly define the ZnTe:Cu layers consistently.

3. Line 37, the bandgap energy of 1.45eV for CdTe semiconductors is just in the proper range for PV with respect to the solar spectrum of AM 1.5G. Thus, the bandgap energy of 1.45eV should not be considered as low bandgap.

4. As the thin-film CdTe solar cells with the superstrate cell configuration have reach the efficiency of 22.1%, justify the merits of CdTe nanocrystal solar cells.

Moreover, please survey the progress and current status of CdTe nanocrystal solar cells.

5. Line 40 and 41, “low ohmic contact”, what does that mean?

6. Line 65, tunnel junction, is it related to the CdTe NC solar cells?

7. Line 68, “there are still few reports on solution-processed CdTe NC-based solar cells.” The references should be cited.

8. Line 72-74, “Inserting a hole-transport layer with high work function before the deposition of an electrode metal contact has been widely accepted for improving the ohmic contact for CdTe NC solar cells.” The references should be cited.

9. Line 86, As the effects of ZnTe:Cu have been studied, do the authors intend to improve Jsc of the cells with ZnTe:Cu layers. However, for the results shown in Table 1, Jsc of the best cell is still less than that of the control cell in this study.

In addition, are the values of Jsc related with the junction quality?

10. Line 93, Do CdTe and CdSe NC work as donor/acceptor materials? Or, are they just p-type and n-type conductivities?

11. Line 96-102, the results shown there should not be given in the section of Introduction. They should be presented and discussed in section 3 of Results and Discussion.

12. Line 104-105, although a reference for the preparation methods is cited. The key process parameters for the deposition of the layers for this study still need to be described in the section of Experiment.

13. Line 108, what is the thickness of each layer for the CdTe NC cells in this study? What is the cell area? In addition, please describe the detailed Characterization methods, including the IV measurements, EQE measurements, and so on.

14. Line 123, what are those RMS values? Roughness?

15. For Fig. 1 b and c, it is not clear that which one has more irregular dots.

16. Line 157, the active layer of the CdTe NC solar cells is merely 600nm. And the short-circuit current densities of these cells are less than 20mA/cm2. Typically the thicknesses of the CdTe layers for thin-film CdTe solar cells are much thicker than 600nm. Are there any particular reasons that the active layers of the CdTe NC solar cells are so thin? Please elaborate.

17. Line 189, the efficiency of the control device without the ZnTe:Cu layer and annealing treatments reached 5.27%. The best cell with the ZnTe:Cu layer annealed at 200 degree C obtained the efficiency of 6.28%. As the best cell was treated with the annealing step and incorporated with the ZnTe:Cu layer, did both the annealing treatments and effects of the ZnTe:Cu layers enhance the efficiency? How to differentiate the impacts of annealing process on the best cell?

In addition, compared with the control device, the open-circuit voltage was significantly improved and fill factor was just slightly improved for the best cell. The short-circuit current density was even slightly decreased. However, in line 195, the authors attribute the performance improvement to the increase in open-circuit voltage and short-circuit current density. The conclusion the authors drew conflict with the results shown in Table 1.

18. Line 196-197, do the authors regard the dark current at the reverse bias as the leakage current? Please provide the references to support this argument. In addition, please provide the references to support the argument that the interface recombination is directly related to the leakage current or dark current.

19. In Table 1, the authors show the effects of the annealing temperatures and the thicknesses of ZnTe layers on the performance parameters of Voc, Jsc, and FF. However, the authors did not provide a thorough discussion about how the annealing temperatures and the thicknesses of the ZnTe layers may impact each of the performance parameters of Voc, Jsc, and FF, and the variation trends of these performance parameters. The manuscript lacks the physical mechanisms explaining these effects.

20. In table 1, the thickness of the Cu layers is only 1 nm. How the thicknesses of the Cu layers were precisely controlled? Furthermore, the thickness of the ZnTe layers are few tens nanometer. How were these thicknesses determined?

21. Line 202-203, the authors claim that the increase in Jsc for the cells with the ZnTe layers agreed well with the improvement of EQE for the whole wavelength. Obviously for the data shown in Table 1, however, the values of Jsc for the cells with the ZnTe layers are less than that of the control cell. Compared with the control cell, the Jsc of the cells with the ZnTe layers decreased. Moreover, the improvement of EQE for the cell with the ZnTe layer is not for the entire range of the wavelengths. The values of Jsc were not estimated from the EQE data either.

22. Line 208-209, “The low EQE response, between 600 to 800 nm, was attributed to the interfacial recombination of CdTe/Au.” But for the EQE shown in Fig. 5(a), the EQE of the CdTe/Au cell is greater than that of CdTe/ZnTe:Cu/Au in the wavelength range of around 660nm to 800nm. Do the result conflict with the statement given in Line 208-209.

23. Line 212-213, “Then, low contact resistance was obtained, which conformed to the experiment results.” Do the authors suggest that the low contact resistance improve the EQE? Why? Please elaborate.

Author Response
list of point-by-point changes based on referee’s report

Nanomaterials-465512
(Italic character- Referee’s comment; red character-authors response to the referee)

Here we submit the revised version of Nanomaterials-465512, in which the comments from the two reviewers are carefully considered and addressed. Our revisions have been highlighted in red color for your easy identification.
Reviewer #
2:
Comments and Suggestions for Authors
The authors mainly studied the effects of ZnTe:Cu layers on the performance of CdTe NC solar cells. The title should be revised in order to better fit the main theme of the study. In addition, the cell configuration shown in Fig. 3(a) for this study is superstrate, which is a common term for thin-film solar cells prepared on glass.
1.
  The abstract should be more concise. The key approaches and the critical results should be presented.
Thanks for the comments. We have revised the abstract according to the referee’s suggestion.

2.
Line 24 and 60, the ZnTe:Cu layers work as the back contacts. In line 96, the ZnTe:Cu layers are regarded as the HTL. In line 33, is the ZnTe:Cu layer also the buffer? Clearly define the ZnTe:Cu layers consistently.
Thanks for the comments. We have defined the ZnTe:Cu as back contact layer and we have revised the manuscript according to the referee’s suggestion.

3. Line 37, the bandgap energy of 1.45eV for CdTe semiconductors is just in the proper range for PV with respect to the solar spectrum of AM 1.5G. Thus, the bandgap energy of 1.45eV should not be considered as low bandgap.

Thanks for the comments. We have defined the bandgap of CdTe NC as moderate bandgap in the revised manuscript. 

4. As the thin-film CdTe solar cells with the superstrate cell configuration have reach the efficiency of 22.1%, justify the merits of CdTe nanocrystal solar cells. Moreover, please survey the progress and current status of CdTe nanocrystal solar cells.

Thanks for the comments. There are many merits of solution processed CdTe NC solar cells, for example, low materials consumed, low cost and simple fabricating technics. Most recently, a new cross-linkable conjugated interface polymer TPA was successfully applied in the CdTe NC-based solar cells, and PCE as high as 8.34% was obtained, which is the highest value ever reported for any solution processed CdTe NCs solar cells with inverted structure[2
9].
5. Line 40 and 41, “low ohmic contact”, what does that mean?
Thanks for the comments. The low ohmic contact means the contact resistance between CdTe and the electrode is low.

6.
Line 65, tunnel junction, is it related to the CdTe NC solar cells?
Thanks for the comments. It is related to the CdTe NC solar cells

7. Line 68, “there are still few reports on solution-processed CdTe NC-based solar cells.” The references should be cited.

Thanks for the comments. We have presented the references in the revised manuscript (reference 2
3).
8.
 Line 72-74, “Inserting a hole-transport layer with high work function before the deposition of an electrode metal contact has been widely accepted for improving the ohmic contact for CdTe NC solar cells.” The references should be cited
Thanks for the comments. We have presented the references in the revised manuscript (reference
9, 25).
9.
Line 86, As the effects of ZnTe:Cu have been studied, do the authors intend to improve Jsc of the cells with ZnTe:Cu layers. However, for the results shown in Table 1, Jsc of the best cell is still less than that of the control cell in this study. In addition, are the values of Jsc related with the junction quality?
Thanks for the comments. The motivation of introducing ZnTe:Cu layers is to facilitate the carriers collecting and improve the whole devices performance (including
Voc, FF, PCE and Jsc). It is noted that Jsc of the best cell is still less than that of the control cell in this study. We speculate that the Cu accumulation in some place of NC solar cells may act as recombination center and decrease the Jsc. Anyway much work should be done to clarify this phenomenon.
The values of
Jsc is related with the junction quality, which is also confirmed in our previous work on solution processed CdTe/CdSe and CdTe/CdS NC solar cells. (reference 31, 32).
10.
Line 93, Do CdTe and CdSe NC work as donor/acceptor materials? Or, are they just p-type and n-type conductivities?
Thanks for the comments. The CdTe and CdSe NC work as donor/acceptor materials
.
11.
Line 96-102, the results shown there should not be given in the section of Introduction. They should be presented and discussed in section 3 of Results and Discussion.
Thanks for the comments. We have revised according to the referee’s suggestion.

12.
Line 104-105, although a reference for the preparation methods is cited. The key process parameters for the deposition of the layers for this study still need to be described in the section of Experiment.
Thanks for the comments. We have described in the section of experiment.

13. Line 108, what is the thickness of each layer for the CdTe NC cells in this study? What is the cell area? In addition, please describe the detailed Characterization methods, including the IV measurements, EQE measurements, and so on.

Thanks for the comments. We have presents all these in the revised manuscript.

14. Line 123, what are those RMS values? Roughness?

Thanks for the comments. The RMS values are the root mean square roughness
(provided by the AFM) of ZnTe thin film with different thickness.
15. For Fig. 1 b and c, it is not clear that which one has more irregular dots.

Thanks for the comments. Fig. 1 b has more irregular dots as the thickness of ZnTe is 10 nm while 20 nm for Fig. 1c. The grain size will be increased as the thickness of ZnTe is increased.

1
6. Line 157, the active layer of the CdTe NC solar cells is merely 600nm. And the short-circuit current densities of these cells are less than 20mA/cm2. Typically the thicknesses of the CdTe layers for thin-film CdTe solar cells are much thicker than 600nm. Are there any particular reasons that the active layers of the CdTe NC solar cells are so thin? Please elaborate.
Thanks for the comments.
 The depletion region for CdTe NC solar cells is ~400 nm, which has been confirmed by Panthani, M. G et al (Reference 25). The optimized thickness of CdTe layers for CdTe NC solar cells with inverted structure is ~500 nm. Device with too thick CdTe layers will suffer from low carrier collecting efficiency and low device performance. In the previous reports, Jsc as high as 23 mA/cm2 is obtained in device structure ITO/ZnO/CdSe/CdTe/Si-TPA/Au, Reference 29.
17. Line 189, the efficiency of the control device without the ZnTe:Cu layer and annealing treatments reached 5.27%. The best cell with the ZnTe:Cu layer annealed at 200 degree C obtained the efficiency of 6.28%. As the best cell was treated with the annealing step and incorporated with the ZnTe:Cu layer, did both the annealing treatments and effects of the ZnTe:Cu layers enhance the efficiency? How to differentiate the impacts of annealing process on the best cell?
In addition, compared with the control device, the open-circuit voltage was significantly improved and fill factor was just slightly improved for the best cell. The short-circuit current density was even slightly decreased. However, in line 195, the authors attribute the performance improvement to the increase in open-circuit voltage and short-circuit current density. The conclusion the authors drew conflict with the results shown in Table 1.

Thanks for the comments. For the solution processed CdTe/CdSe NC solar cells, the optimized annealing treatments parameters has been set up in the previous reports (
reference 31). There are almost no change in NC solar cells performance when further annealing under 200 ℃ (device structure ITO/ZnO/CdSe/CdTe/Au). Without annealing (device structure ITO/ZnO/CdSe/CdTe/ZnTe:Cu/Au), the PCE of NC solar cells is only 2.45%, as presented in the following. After annealing, the grain size of ZnTe will increase and Cu will diffuse into the whole ZnTe thin film, therefore, homogeneous and high quality ZnTe:Cu will be obtained after annealing.
The performance improvement is attributed to the increase in open-circuit voltage and fill factor. We have revised according to the referees’ suggestion. 

18. Line 196-197, do the authors regard the dark current at the reverse bias as the leakage current? Please provide the references to support this argument. In addition, please provide the references to support the argument that the interface recombination is directly related to the leakage current or dark current.

Thanks for the comments. The dark current at the reverse bias is not the leakage current. It refers to the current measured under dark conditions at reverse bias. The low current in reverse bias voltage under dark and improved
 Rsh imply that the ZnTe:Cu can serve as electron blocking layer to effectively prevent the leakage currents (reference 40).
1
9. In Table 1, the authors show the effects of the annealing temperatures and the thicknesses of ZnTe layers on the performance parameters of Voc, Jsc, and FF. However, the authors did not provide a thorough discussion about how the annealing temperatures and the thicknesses of the ZnTe layers may impact each of the performance parameters of Voc, Jsc, and FF, and the variation trends of these performance parameters. The manuscript lacks the physical mechanisms explaining these effects.
Thanks for the comments. We have provided discussion about the effects of annealing temperatures and the thicknesses of the ZnTe layers on the device performance in the revised manuscript.

20.
In table 1, the thickness of the Cu layers is only 1 nm. How the thicknesses of the Cu layers were precisely controlled? Furthermore, the thickness of the ZnTe layers are few tens nanometer. How were these thicknesses determined?
Thanks for the comments. The thickness of Cu and ZnTe layers are determined by the quartz balance equipped in the evaporating instrument. The thickness is further calibrated by a Profilometer.

21. Line 202-203, the authors claim that the increase in Jsc for the cells with the ZnTe layers agreed well with the improvement of EQE for the whole wavelength. Obviously for the data shown in Table 1, however, the values of Jsc for the cells with the ZnTe layers are less than that of the control cell. Compared with the control cell, the Jsc of the cells with the ZnTe layers decreased. Moreover, the improvement of EQE for the cell with the ZnTe layer is not for the entire range of the wavelengths. The values of Jsc were not estimated from the EQE data either.

Thanks for the comments. We have revised the manuscript according to the referee’s comments. The values of Jsc were estimated from the EQE data.

22. Line 208-209, “The low EQE response, between 600 to 800 nm, was attributed to the interfacial recombination of CdTe/Au.” But for the EQE shown in Fig. 5(a), the EQE of the CdTe/Au cell is greater than that of CdTe/ZnTe:Cu/Au in the wavelength range of around 660nm to 800nm. Do the result conflict with the statement given in Line 208-209.

Thanks for the comments. We have revised this part in the manuscript.

23. Line 212-213, “Then, low contact resistance was obtained, which conformed to the experiment results.” Do the authors suggest that the low contact resistance improve the EQE? Why? Please elaborate.

Thanks for the comments. The low contact resistance couldn’t improve the EQE. We have revised the expression in the manuscript.

Round  2

Reviewer 2 Report

1. As the authors have defined the ZnTe:Cu layers as the back contacts, why does the term of the buffer layer still appear in line 32?

2. As the authors described that “there are still few reports on solution-processed CdTe NC-based solar cells” in line 67 and 68, some more references should be cited accordingly.

3. In line 41, the description of “low ohmic contact” is ambiguous. Please describe clearly.

4. Given the exact meaning of the tunnel junction, specify that which parts of the cells or devices play the role of the tunnel junction in lie 65, and the function of the tunnel junction to the cells or devices.

5. In line 89 and 90, how was the junction quality of the solar cells determined? How to know the junction quality is good or poor?

6. For the AFM images given in Figure 1, which parts can be identified as the irregular dots on the image?

Author Response

List of point-by-point changes based on referee’s report

Nanomaterials-465512

(Italic character- Referee’s comment; red character-author’s response to the referee)
Here we submit the revised version of Nanomaterials-465512, in which the comments from the reviewer is carefully considered and addressed. Our revisions have been highlighted in red color for your easy identification.

Reviewer #2:

Comments and Suggestions for Authors
1.
 As the authors have defined the ZnTe:Cu layers as the back contacts, why does the term of the buffer layer still appear in line 32?
Thanks for the comments. We have replaced the buffer layer as ZnTe:Cu according to the referee’s suggestion.

2. As the authors described that “there are still few reports on solution-processed CdTe NC-based solar cells” in line 67 and 68, some more references should be cited accordingly.

Thanks for the comments. We have added more references in the revised manuscript.

3.  In line 41, the description of “low ohmic contact” is ambiguous. Please describe clearly.

Thanks for the comments. The low ohmic contact means the low resistance of CdTe/back contact electrode. We have described in the revised manuscript.

4. Given the exact meaning of the tunnel junction, specify that which parts of the cells or devices play the role of the tunnel junction in lie 65, and the function of the tunnel junction to the cells or devices.

Thanks for the comments. The ZnTe:Cu parts of the cells play the role of the tunnel junction.
Furthermore, the ZnTe:Cu layer acts as electron block layer and hole transfer layer for hole collecting.
5.In line 89 and 90, how was the junction quality of the solar cells determined? How to know the junction quality is good or poor?

Thanks for the comments. When the solar cells consist of the same device structure, device with low leakage current means high junction quality. Usually, device with high junction quality shows higher fill factor and parallel resistance.  

6. For the AFM images given in Figure 1, which parts can be identified as the irregular dots on the image?

Thanks for the comments. The irregular dots here is referred as white dots in the Figure 1b .
 In order to clarify, we have modified the irregular dots as particle.  
